# Tyrosine Kinase Inhibitors in the Treatment of Metastasised Renal Cell Carcinoma—Future or the Past?

**DOI:** 10.3390/cancers14153777

**Published:** 2022-08-03

**Authors:** Jakob Michaelis, Markus Grabbert, August Sigle, Mehmet Yilmaz, Daniel Schlager, Christian Gratzke, Arkadiusz Miernik, Dominik Stefan Schoeb

**Affiliations:** Department of Urology, Faculty of Medicine, Medical Centre, University of Freiburg, Hugstetter Str. 55, D-79106 Freiburg, Germany; jakob.michaelis@uniklinik-freiburg.de (J.M.); markus.grabbert@uniklinik-freiburg.de (M.G.); august.sigle@uniklinik-freiburg.de (A.S.); mehmet.yilmaz@uniklinik-freiburg.de (M.Y.); daniel.schlager@uniklinik-freiburg.de (D.S.); christian.gratzke@uniklinik-freiburg.de (C.G.); arkadiusz.miernik@uniklinik-freiburg.de (A.M.)

**Keywords:** tyrosine kinase inhibitors, adjuvant therapy, immune checkpoint inhibitors, renal cell carcinoma, targeted therapy

## Abstract

**Simple Summary:**

Renal cell carcinoma (RCC) is the sixth most frequently diagnosed cancer in men and the tenth in women with a rising incidence. The treatment of metastasized RCC has dramatically changed in the last decade, improving the overall survival of patients significantly. In this context, cornerstones of the treatment have been tyrosine kinase inhibitors (TKI), with Sunitinib being the preferred first-line treatment for most cases. With the introduction of immunotherapy and combination therapy, this changed recently. The current article summarizes the available literature on TKI treatment of metastasized RCC and shows the current part of TKIs in the treatment algorithm as well as its potential future role.

**Abstract:**

Background: To review and discuss the literature on applying tyrosine kinase inhibitors (TKIs) in the treatment of metastasised renal cell carcinoma (mRCC). Materials and Methods: Medline, PubMed, the Cochrane database, and Embase were screened for randomised controlled trials, clinical trials, and reviews on treating renal cell carcinoma, and the role of TKI. Each substance’s results were summarised descriptively. Results: While TKI monotherapy is not currently recommended as a first-line treatment for metastasized renal cell carcinoma, TKIs are regularly applied to treat treatment-naïve patients in combination with immunotherapy. TKIs depict the first-choice alternative therapy if immunotherapy is not tolerated or inapplicable. Currently, seven different TKIs are available to treat mRCC. Conclusions: The importance of TKIs in a monotherapeutic approach has declined in the past few years. The current trend toward combination therapy for mRCC, however, includes TKIs as one significant component of treatment regimens. We found that to remain applicable to ongoing studies, both when including new substances and when testing novel combinations of established drugs. TKIs are of major importance for the treatment of renal cancer now, as well as for the foreseeable future.

## 1. Introduction

Renal cell carcinoma (RCC) is the sixth most frequently diagnosed cancer in men and the tenth in women. Its incidence has been rising for the last several years, at least partly due to the more accurate imaging modalities detecting even very small masses. Even though this improvement and thus earlier detection often enable curative surgical resection, approximately 17% of patients [1] will harbour a metastasis and require systemic medical therapy. The treatment landscape for metastasised renal cell carcinoma (mRCC) has changed fundamentally over the last two decades, and new targeted therapies have significantly improved the prognosis of these patients. Among these therapies, the tyrosine kinase inhibitors (TKI) sorafenib and sunitinib were the first novel therapies approved for advanced RCC in Europe in 2006. Since then, many more TKIs have been introduced, but guideline recommendations of when to use them have also changed significantly since the introduction of immunotherapeutic agents (IO). While initially used as a monotherapeutic agent and first-line treatment, TKIs are now usually administered together with immunotherapy. TKI monotherapy is nowadays preferred in later therapy lines or in patients that are ineligible for IO. Most recent studies investigated the effects, interactions, and sequencing of various therapies containing TKIs [2,3,4,5]. However, the wide variety of available substances makes it challenging for the clinician to choose which substance to apply in a specific case. The present article provides an overview of the currently available TKIs for treating mRCC, and the existing data on how and when to apply them. 

## 2. Materials and Methods

We conducted a literature search using predefined Medical Subject Headings (MeSH) terms on Medline, PubMed, the Cochrane database, and Embase to identify related randomised controlled trials, clinical trials, and reviews on treating renal cell carcinoma using TKIs, performing an individual search for each substance. Publications relevant to the subject and their cited references were retrieved and appraised independently by two authors (J.M. and D.S.S). In addition, we screened clinical trial databases for current trials involving TKI treatment for renal cancer as well. Systematic reviews and clinical studies (randomised controlled trials, cohort studies, case-control studies, and case series) were included. In addition, information on animal studies, non-systematic reviews, and publications with ‘Epub ahead of print’ status were also included. Non-English-language articles, case reports, publications based on expert opinion, physiology/bench research or ‘first principles’, epidemiological studies, cross-sectional studies, and cadaveric studies were excluded. Two authors (J.M. and D.S.S.) extracted data from the selected publications, including study characteristics, information about the intervention, patient characteristics, and treatment outcomes. Extracted data were then evaluated by all participating authors and are presented descriptively in this manuscript. 

## 3. Results

### 3.1. General Mechanism of TKI Therapeutic Effect

Protein tyrosine kinases are a family of proteins playing a key role in numerous signalling pathways affecting cell growth, cell differentiation, and metabolism. They are divided broadly into the categories of receptor vs. nonreceptor tyrosine kinases, and both categories are involved in the developmental pathway of multiple cancer types [2]. Because of their oncogenic potential, tyrosine kinases have become a target for directed therapy in cancer. In RCC treatment, the main therapeutic target in this context is the vascular endothelial growth factor (VEGF) receptor family, which consists of three membrane receptor tyrosine kinases. Activated by VEGF, a tumour-secreted cytokine, the VEGF receptor promotes tumour growth by inducing angiogenesis. In the pathogenesis of RCC, inactivation of both alleles in the von Hippel–Lindau (VHL) gene causes hypoxia-induced factors (HIF) to over-accumulate. The simulation of a hypoxic state in the cell leads to the transcription of VEGF and several other pro-angiogenic factors [3]. Antiangiogenic TKIs can obstruct this pathway by inhibiting the VEGF-receptor kinases and have an advantage over, e.g., monoclonal antibodies, by being orally bioavailable. Additionally, the substances applied in the clinical treatment of RCC are so-called multikinase or multitargeted tyrosine kinase inhibitors, which have specificity to a broader spectrum of tyrosine kinases due to their resemblance to the ATP binding site region. This enables them to inhibit multiple signalling pathways involving other tyrosine kinase receptors that can contribute to tumour growth. Examples are the fibroblast growth factor receptor (FGFR), c-kit receptor, platelet-derived growth factor receptor, and epidermal growth factor receptor [4]. 

New-generation TKIs stand out by addressing a broader spectrum of tyrosine kinases (e.g., lenvatinib or cabozantinib) or by being more potent, which is objectified by lower half maximal inhibitory concentrations/IC_50_ (e.g., axitinib or tivozanib) [PMID 29033542]. While targeting multiple receptor tyrosine kinases aims at preventing resistance, higher potency or selectivity bears the potential to reduce side effects. 

Acquired resistance mechanisms against TKIs include the activation of VEGF-independent “bypass pathways”, sustaining or reinducing angiogenesis. Various studies revealed numerous contributors to resistance, such as lysosomal sequestration of, e.g., sunitinib (PMID 21980135), proangiogenic inflammatory cell recruitment (PMID 30447930), or epithelial–mesenchymal transition (PMID 26940073). These mechanisms can be addressed.

### 3.2. Characteristics and Trial Evidence for Currently Available TKI

#### 3.2.1. Sunitinib

As mentioned above, Sunitinib was among the first TKIs approved to treat mRCC. Its comparison with the previous gold standard at that time (Interferon-α treatment) revealed a progression-free survival (PFS) benefit for Sunitinib of 6 months [5]. In the first decade of Sunitinib therapy, the overall survival of mRCC more than doubled, not in small part due to the new TKI treatment [6]. Furthermore, the establishment of Sunitinib as a reference standard for the treatment of mRCC with previously unseen response rates has improved our understanding of TKI treatment and the management of side effects in general, which played a key role in the successful development of new-generation TKIs, as well as the clinical implementation of new combination therapies [7].

Sunitinib as a first-line treatment has since been replaced. In several trials, Sunitinib served as the comparator against combination therapy, and was found to be inferior [8,9,10,11]. For example, in the CheckMate 9ER [8] trial, a combination of Cabozantinib and Nivolumab demonstrated superior overall survival (OS), as well as PFS, and a more likely response than Sunitinib monotherapy. Additionally, adjuvant treatment with Sunitinib after cytoreductive surgery has been investigated. Only one randomised clinical trial [12] showed a benefit in PFS in high-risk patients treated with Sunitinib in an adjuvant setting, without, however, revealing any significant influence on OS, but a higher risk for toxic effects. Current guidelines therefore do not recommend adjuvant treatment with Sunitinib [13]. 

#### 3.2.2. Sorafenib

Sorafenib inhibits angiogenesis and tumour progression by limiting the effects of various receptor tyrosine kinases such as VEGFR-2, VEGFR-3, c-KIT, or platelet-derived growth factor receptor beta (PDGFRβ) [14]. Between 2003 and 2005, the phase III randomised TARGET trial tested sorafenib against a placebo for metastatic and/or unresectable RCC as a second-line treatment after IFNα. In total, 903 patients were 1:1 randomized, and the primary endpoint was OS. Median PFS showed sorafenib’s superiority with 5.5 vs. 2.8 months in the placebo group (hazard ratio (HR) 0.44, *p* < 0.01), resulting in crossover permission. An interim analysis prior to crossover demonstrated OS favouring sorafenib, as shown by an HR of 0.72 (*p* = 0.02) when compared to placebo therapy [15]. Comparable results in PFS terms were achieved in subsequent studies, expanding sorafenib’s use to in non-clear-cell RCC and first-line treatment [16,17,18]. While the complete response was scarce with <1%, 60% to 80% of study participants initially achieved stable disease [17,18]. Nevertheless, there were many adverse events (AEs) and severe adverse events (SAEs, defined as AEs ≥ grade 3), in particular hand–foot skin reactions, hypertension, haemorrhagic complications, diarrhoea, and fatigue [18,19].

Investigating the optimal sequential treatment, the CROSS-J-RCC and SWITCH trials detected no significant difference between the sunitinib/sorafenib and sorafenib/sunitinib sequences for first- and second-line therapy [20,21]. As with sunitinib, sorafenib served as a comparator in several studies assessing novel treatment options. Compared to tivozanib in the TIVO-1 trial, clear-cell mRCC patients had worse PFS under sorafenib therapy, but similar OS [22]. In particular, sorafenib appeared to be an ineffective first-line treatment option in the MSKCC favourable risk group, since this subgroup’s outcome in TIVO-1 was poor, and sorafenib/sunitinib sequential therapy was inferior to sunitinib/sorafenib in the CROSS-J-RCC study. The AXIS trial demonstrated that axitinib yielded better PFS after previous VEGF-inhibitor therapy (mostly sunitinib) with 8.3 vs. 5.7 months [23]. Hence, now that IO constitutes the predominant first-line treatment, other VEGF-inhibitors such as tivozanib have revealed PFS superiority against sorafenib and axitinib, thus representing a reasonable third-line option after VEGF-inhibitors. Additionally, sorafenib failed to demonstrate any beneficial effect [24,25] in an adjuvant setting when compared to the placebo. In summary, sorafenib no longer plays a substantial role in current RCC treatment.

#### 3.2.3. Cabozantinib

Cabozantinib inhibits many tyrosine kinase receptors, such as ET, VEGFR-1-3, KIT, TRKB, FLT-3, and TIE-2. Moreover, in contrast to other TKIs, it has relevant activity against MET and AXL. In a randomised multicentre trial (The Alliance A031203 CABOSUN Trial), monotherapy with 60 mg of cabozantinib daily compared to sunitinib standard therapy (50 mg once per day; 4 weeks on, 2 weeks off) led to a significant PFS benefit (8.2 vs. 5.6 months) and an increased objective response rate (ORR) (33% vs. 12%). The safety profiles of both drugs did not differ significantly, with an overall incidence rate of grade 3 or 4 AE in 67% of those receiving cabozantinib and 68% receiving sunitinib [26]. Additionally, the open-label, randomised phase 3 trial (METEOR) compared cabozantinib with everolimus to treat clear-cell mRCC and showed better OS (median 21.4 months vs. 16.5) and ORR (17% [8,9,10,11,12,13,14,15,16,17] vs. 3% [2,27,28,29,30]) with cabozantinib 60mg/daily compared to Everolimus 10mg/daily. Included in this trial were patients who had already undergone treatment with one or more VEGFR TKI [31]. The SWOG 1500 trial included patients with metastatic papillary RCC [33592176] in a randomised four-arm setting testing cabozantinib, sunitinib, savolitinib, and crizotinib. Cabozantinib proved to be superior to sunitinib with a median PFS of 9.0 months vs. 5.6 months (HR 0.60, *p* = 0.02), establishing it as the front-line therapeutic option for this histological subtype. Median OS showed no significant differences. The crizotinib and savolitinib study arms (mPFS 2.8 and 3.0 months, respectively) were closed prematurely because of their inefficacy. 

Combined with Nivolumab (240mg/biweekly), Cabozantinb 40mg was compared in a randomised, open-label trial with sunitinib (50 mg once daily for 4 weeks in each 6-week cycle) monotherapy in patients with previously untreated clear-cell advanced RCC (CheckMate 9ER trial) [8]. The IO/TKI combination revealed both a better median PFS (16.6 vs. 8.3 months, HR 0.51, *p* < 0.001) and OS (HR 0.60, *p* = 0.001) than Sunitinib [PMID 33657295]. Patients also reported a higher health-related quality of life in the IO/TKI combination study group. However, grade 3 or higher adverse events occurred in 75.3% of study-group patients compared to 70.6% in the control group. ORR for Nivolumab/Cabozantinib resembled ORR results from trials for Pembrolizumab/Axitinib or Avelumab/Axitinib (55.7% vs. 59.3% and 56.0%, respectively) while the median PFS even seemed to be superior for Nivolumab/Cabozantinib. This is noteworthy given the fact that the Checkmate 9ER study population contained the highest proportion of IMDC poor-risk patients, representing a potential restrictor to outcome parameters. 

As a consequence, the latest treatment guidelines for mRCC recommend Cabozantinib (40 mg/daily) in combination with Nivolumab as first-line treatment and as monotherapy (60mg/daily) as second-line treatment, especially as an alternative therapeutic approach if combination therapy had to be discontinued because of immune therapy-related side effects. Moreover, the European Association of Urology (EAU) 2021 guidelines advise administering cabozantinib for metastatic papillary RCC without further molecular testing.

#### 3.2.4. Axitinib

As a comparatively selective TKI, axitinib already inhibits VEGF1-3 at significantly lower concentrations (below nanomolar) than other TKIs [32]. In the randomised, multicentric, phase-3 AXIS trial, axitinib was compared to sorafenib as second-line therapy in 723 patients with mRCC [23]. Here, median PFS was significantly longer compared to sorafenib (8.3 months vs. 5.7 months, HR= 0.66, CI 0.55-0.78, *p* < 0.0001). Axitinib also stood out through its less-severe spectrum of side effects, evident in fewer AE-related treatment discontinuations (4% vs. 8%) [33]. Diarrhoea (55%), hypertension (40%), and fatigue (39%) were the most frequent AEs in the axitinib arm, with hypertension representing the predominant SAE; on the other hand, sorafenib side effects were characterized by diarrhoea (53%), alopecia (32%), and palmar-plantar erythrodysaesthesia (51%), which was the most common SAE. In contrast to that, axitinib failed to yield longer PFS as a first-line treatment than sorafenib (10.1 months vs. 6.5 months, HR 0.77, CI 0.56–1.05, one-sided *p* = 0.04) [34], although this study might be underpowered on account of the previously underestimated efficacy of the comparator sorafenib. 

In the multicentre, open-label, single-arm, phase II AXIPAP trial, axitinib’s efficacy was investigated in 42 patients with advanced or metastatic papillary RCC [35]. Their overall median PFS, median PFS for type 1 papillary RCC, and median PFS for type 2 papillary RCC were 6.6 months (95% CI, 5.5–9.2), 6.7 months (95% CI, 5.5–9.2), and 6.2 months (95% CI, 5.4–9.2), respectively. Type 2 papillary RCC showed a rather high 36% ORR though the differences did not reach statistical significance because of the small cohort size. The median overall OS was 18.9 months. 

In KEYNOTE-426, the IO/TKI combination of pembrolizumab and axitinib showed a superior median PFS (15.1 vs. 11.1 months, HR 0.69, *p* < 0.001) and ORR (59.3% vs. 35.7%, *p* < 0.001) compared to sunitinib, which led to better OS (HR 0.53, *p* < 0.0001) for combination therapy [9] as well. Based on these data, pembrolizumab/axitinib is currently recommended as a first-line mRCC treatment. The JAVELIN RENAL 101 trial investigated axitinib with avelumab against sunitinib, displaying advantageous median PFS (13.8 vs. 7.0 months, HR 0.62, *p* < 0.001), but lacking proof of a subsequent OS benefit (HR 0.83, *p* = 0.13) for PD-L1 positive tumours [36]. 

#### 3.2.5. Tivozanib 

Tivozanib, similar to axitinib, stands out as a specific VEGFR 1–3 inhibitor with only a few additional inhibitions of, e.g., PDGFR-β und c-Kit [37]. Tivozanib was introduced as an mRCC therapy option through the aforementioned phase III RCT TIVO-1 against sorafenib in VEGF- (and mTOR-) treatment-naïve patients, showing tivozanib’s longer median PFS (11.9 vs. 9.1 months, HR 0.797, *p* = 0.042) [22]. Tivozanib was also tested as a third-line treatment after previous TKI therapy against sorafenib and showed a better median PFS of 5.6 months compared to 3.9 months under sorafenib (*p* = 0.02) [38]. Noteworthily, there were no differences in SAEs, but a different spectrum of side effects. Tivozanib treatment was associated with higher rates of hypertension and dysphonia, while sorafenib was—in line with the trials mentioned above—associated with hand–foot skin reactions, diarrhoea, and rash. 

#### 3.2.6. Lenvatinib

Lenvatinib targets multiple receptor tyrosine kinases such as VEGFR1-3, PDGFRα, KIT, RET, and FGFR [39]. Lenvatinib can be given as a combination therapy with everolimus based on a three-armed study’s results in which lenvatinib, everolimus, and their combination were tested as second-line treatment after progression under prior TKI monotherapy. This randomised phase-II trial showed longer PFS for lenvatinib/everolimus (HR 0.40, CI 0.24–0.68, *p* < 0.01) and single-agent lenvatinib (HR 0.61, CI 0.38–0.98, *p* = 0.048), when compared to everolimus, respectively. Combination lenvatinib/everolimus therapy yielded the longest median PFS with 14.6 months, although its superiority over lenvatinib monotherapy was not proven (PFS 7.4 months, HR 0.66, CI 0.30–1.10, *p* = 0.12) [40]. Lenvatinib had poor outcomes concerning treatment side effects, as the SAEs were more frequent in those treatment arms that included lenvatinib, with SAEs evident in 71% and 79% of those receiving lenvatinib combination- and single-agent-therapy, respectively. As a special feature, proteinuria was the most common SAE in conjunction with lenvatinib monotherapy (19%), followed by hypertension and diarrhoea. Constipation, diarrhoea, and hypertension characterised the SAE profile of combination therapy. 

Lenvatinib paired with pembrolizumab is the most recent combination therapy for mRCC. In the three-armed CLEAR study, the combinations of lenvatinib/pembrolizumab and lenvatinib/everolimus were tested against sunitinib, and both yielded longer mPFS (23.9 vs. 9.2 months, HR 0.39, *p* < 0.001 and mPFS 14.7 vs. 9.2 months, HR 0.65, *p* < 0.001, respectively), but only lenvatinib/pembrolizumab proved to be superior in OS terms (HR 0.66, *p* = 0.005) [41]. Lenvatinib/pembrolizumab therapy stands out for yielding the highest reported ORR (71.0%) of all available therapies, and the highest rate of complete responses (16.1%). On the other hand, this study included fewer patients in the poor and more participants in the favourable IMDC prognosis group than trials investigating other IO/TKI combination therapies, thus limiting their comparability and potentially contributing to their superior efficacy data. Beyond that, the lenvatinib/pembrolizumab combination’s toxicity stands out with AE ≥ grade 3 of 82.4% (compared to 75.8% for pembrolizumab/axitinib or 75.3% for nivolumab/cabozantinib), leading to the highest proportion of treatment discontinuation due to AEs at 37.2% (compared to pembrolizumab/axitinib’s 30.5% and nivolumab/cabozantinib’s 19.7%). Nonetheless, health-related quality of life was not worse than with sunitinib [10]. 

#### 3.2.7. Pazopanib

Pazopanib inhibits VEGFR1-3, PDGFR alpha and beta, FGFR1 and 3, c-kit, and other tyrosine kinases [42]. Pazopanib was first tested against a placebo in a randomised, double-blind, placebo-controlled phase III study in 435 patients with locally advanced and/or mRCC. Median PFS was shown to be significantly prolonged with pazopanib compared to the placebo in the general study population (9.2 vs. 4.2 months; HR = 0.46, *p* < 0.001), regardless of whether patients lacked previous treatment (11.1 months vs. 2.8 months; HR = 0.40, *p* < 0.0001) or were cytokine-pretreated (7.4 months vs. 4.2 months; HR = 0.54, *p* < 0.001) [43]. In addition, the pazopanib arm’s ORR was significantly higher (30% vs. 3%, *p* < 0.001). Diarrhoea, hypertension, hair colour changes, and elevated ALAT/ASAT were the most common AEs. The phase-III COMPARZ trial compared pazopanib and sunitinib as clear-cell mRCC first-line therapy: Pazopanib’s non-inferiority was proven through its median PFS compared to sunitinib (8.4 months for pazopanib and 9.5 months for sunitinib) [44]. OS was similar between these groups (28.4 vs. 29.3 months, HR = 0.91; *p* = 0.28). Note that the pazopanib group’s quality of life seemed better, mainly due to lower levels of fatigue and less mouth, throat, hands, and feet soreness. By that, pazopanib use was linked to significantly greater satisfaction with the therapy compared to sunitinib [44]. This finding is consistent with the PISCES trial results, which featured an intended sequential cross-over design with these two drugs; they found pazopanib to be patient-preferred, primarily thanks to better QoL and less fatigue [45]. A single-centre study investigated pazopanib in comparison to sunitinib for the subset of poor-risk mRCC (defined according to the ARCC trial [46]), in which—in contrast to the COMPARZ trial—pazopanib proved to yield longer PFS and OS than sunitinib [47].

#### 3.2.8. Other TKI—Anlotinib and Savolitinib

Although they are not referred to in most guidelines, TKIs other than the above-mentioned have been tested with promising results in smaller phase-II and phase-III studies. For example, there is limited efficacy evidence on anlotinib, an inhibitor of VEGFR, PDGFR, and c-Kit [48]. As a second-line option after progression under (or intolerance to) sunitinib or sorafenib, anlotinib treatment achieved a median PFS and OS of 14.0 and 21.4 months in 42 patients, respectively. The subgroup suffering disease progression that led to anlotinib therapy revealed a worse outcome, namely a median PFS of 8.5 months and OS of 20.4 months. The Aes resembled those associated with other TKIs, with diarrhoea, hypertension, and hand–foot skin reactions being most common [49]. Another phase-II study examined anlotinib’s effectiveness vs. sunitinib’s as first-line therapy for mRCC [50] in a 2:1 randomisation setting. Median PFS and OS did not differ significantly between groups (*n* = 90 vs. *n* = 43) with 17.5 and 30.9 months for anlotinib and 16.6 and 30.5 months for sunitinib, respectively. Anlotinib’s AE characteristics resembled the above-mentioned second-line study. Note that SAEs were less common in conjunction with anlotinib than sunitinib with 28.9 vs. 55.8%. Nevertheless, health-related quality of life was similar between treatment groups. With this fact as the key negative outcome vs. the COMPARZ trial, and lacking evidence-based on phase-III trials, Anlotinib is currently not recommended as mRCC treatment.

Savolitinib, a highly specific MET inhibitor [51], gained interest as a therapeutic approach for papillary RCC, where MET-gene alterations on chromosome 7 represent a crucial factor in tumorigenesis [52]. In the phase-III SAVOIR trial, savolitinib was investigated in MET-driven metastatic papillary RCC compared to sunitinib [53]. The trial had difficulty recruiting patients, randomising only 60 (33 vs. 27) of 180 preplanned patients in over two years. Median PFS did not differ significantly, with 7.0 months for savolitinib and 5.6 months for sunitinib (HR 0.71, *p* = 0.31), but savolitinib was associated with less toxicity with Aes ≥ grade 3 in 42% vs. 81% for sunitinib. With disappointing results from the SWOG 1500 trial’s cohort of metastatic papillary RCC, savolitinib should only be administered in approved MET-driven papillary RCC, according to the EAU 2021 guidelines on RCC. 

### 3.3. Role of TKIs in Adjuvant RCC Therapy

Although there have been five large, randomised trials investigating adjuvant therapy in locally and locally advanced diseases, their results have been conflicting [54,55]. Sunitinib was linked to limited disease-free survival (DFS) benefit with a median DFS of 6.8 vs. 5.6 years (*p* = 0.03) against a placebo in one trial [53], while another study found no significant differences in DFS [56]. Furthermore, axitinib was tested in locoregional renal cell carcinoma after nephrectomy against a placebo in the ATLAS trial. Axitinib’s utility failed to be proven in the overall population, but it yielded improved DFS [57]—exclusively in the subgroup of highest-risk patients (defined as pT3 with Fuhrman grade 3 or higher, pT4 or pN+). Pazopanib yielded DFS advantages at higher doses of 800 mg daily against placebo in the PROTECT trial. However, the dosage had to be reduced because of high AE-related discontinuation rates, and the DFS benefits disappeared for adjusted pazopanib dosage. Beyond that, OS was similar between groups irrespective of the pazopanib dosage [58].

Summing up, the EAU 2021 guidelines on RCC state that adjuvant sorafenib, pazopanib, or axitinib do not improve DFS and OS after nephrectomy (level of evidence, LE = 1b) and are therefore not recommended as adjuvant therapy. In addition, there is no evidence that TKI treatment prolongs recurrence-free survival (RFS) in patients with no evidence of residual disease after metastasectomy (LE = 1b), hence TKI treatment is not recommended in this subgroup of patients either [59]. For detailed results from the trials investigating adjuvant therapy with TKIs, see Table 1.

As the Keynote-564 study group reported positive results for pembrolizumab [60] and TKIs failed to lengthen DFS, consecutive trials for adjuvant therapy of renal cell carcinoma left TKI-based therapy approaches. By that, future prospects for adjuvant therapy are directed more toward immunotherapies, with results from trials being awaited that are investigating immune checkpoint inhibitors as monotherapy (atezolizumab, NCT03024996) as well as in combination with CTLA-4-inhibitors (Durvalumab + Tremelimumab, NCT03288532; Nivolumab + Ipilimumab, NCT03138512) or HIF-2α inhibitors (Pembrolizumab + Belzutifan, NCT05239728). 

## 4. Discussion and Conclusions—Present and Future Developments for the Treatment of mRCC Involving TKI Therapy

The therapeutic landscape for mRCC has noticeably broadened in recent years. Keeping track of the various studies in the field of mRCC may pose a challenge for urologists making therapy decisions. Hence, giving an overview and prioritizing the available data were our main motives for this review. 

Combination therapy including TKI is the front-line therapy choice for mRCC, containing axitinib [36,61], cabozantinib [8], or lenvatinib [41]. The preference for TKI-containing combination therapy is linked to supposed synergistic effects by influencing the tumour microenvironment and causing imunogenic modulation [62], which might enhance the therapeutic efficacy of immune checkpoint inhibitors. Note that the only widely recommended first-line option without including a TKI is combined nivolumab/ipilimumab [11]. Meanwhile, the present guidelines no longer list TKI monotherapy as the first choice as initial mRCC treatment because of the superiority of TKI/IO or IO/IO combination therapies in OS and PFS terms [1]. Considering all these trials, TKI monotherapy failed to achieve either mPFS exceeding 1 year or a 35% ORR. IO/TKI combination therapy clearly surpassed these benchmarks in several trials [36,41,61], thereby becoming established as the most widely accepted first-line therapy approach. For a timeline of the development of the medical therapy of metastasized RCC, see Figure 1.

TKI monotherapy is currently administered as an alternative strategy as first-line treatment if the patient cannot tolerate or undergo immunotherapy. As monotherapy, TKI should be preferred over mTOR inhibitors according to the trial evidence [40,63]. In the case of contraindication for IO, the TKI/mTOR-I combination of lenvatinib/everolimus is the most efficacious therapy regarding PFS [40], but failed to reveal an OS benefit compared to sunitinib in the CLEAR trial [41]. 

TKI monotherapy is recommended in a second-line therapeutic setting after progression under IO/IO or TKI/IO therapy. Depending on the TKI used, it is advisable to switch to another substance in the second-line setting. Second-line TKI treatment after TKI monotherapy can also be applied if combination therapy is unsuitable. Cabozantinib is recommended in both cases as the first choice provided it was not previously administered. Alternatively, axitinib can be given as well. 

Regarding the IO/TKI trials leading to first-line approval, approximately 15 to 20 percent of patients starting with IO/TKI combination therapy must discontinue IO treatment because of therapy-related adverse events but are able to continue the already-established TKI as monotherapy. Adding patients with contraindications for IO or with a preference for TKI (e.g., because of oral intake), a relevant proportion of patients still undergoes TKI monotherapy early, without disease progression under treatment leading to it. 

Beyond these factors, note that study populations are carefully selected, and they cannot accurately depict real-world patient cohorts. The fact that many patients with mRCC are ineligible for clinical trials because of their comorbidities [64] represents an element of uncertainty for treatment recommendation by possible selection bias. Having access to more real-world data on TKIs might back up TKI therapy choices for patients not represented by study cohorts. In this context, for example, proof of pazopanib’s effectiveness exists from the observational PRINCIPAL study, showing that patients experienced similar mPFS and mOS, regardless of their eligibility for clinical trials [64]. There is a paucity of such observational data on combination treatments containing IO. 

It is debatable how much low performance scores and comorbidities, both linked to already limited life expectancy, should shift the focus of rationale for treatment selection from oncologic efficacy to the toxicity profile or health-related quality of life. Considering the latter, pazopanib yielded superior outcomes regarding treatment side effects or patient-reported satisfaction with therapy and was patient-preferred compared to sunitinib [44]. Pazopanib may therefore constitute a recommendable treatment option for selected patients, particularly those not represented in clinical trials due to age, a low performance score, or comorbidities. 

Age should not be a primary decision criterion for whether or not a patient should obtain IO-containing combination therapy. Nevertheless, it should be considered that IO-based therapies yielded worse ORR and OS in patients ≥70 years old compared to younger patients [65], while TKIs seem to have consistent efficacy in both older patients and in younger ones [64,66]. 

The major limitations of this work lie in the non-predefined mode of selecting information about the discussed substances, which poses the risk of biased, subjective selection of what is to be highlighted. The authors are influenced by their individual experiences with each therapy, which may lead to availability bias becoming a major factor in selecting characteristics to be presented. 

TKI will likely continue to be a key component in combination therapy approaches, since new IO/TKI combinations are being investigated in current RCC trials, such as nivolumab/ipilimumab/cabozantinib (NCT03937219, NCT03793166), durvalumab/savolitinib for MET-driven papillary RCC (NCT05043090), or toripalimab/axitinib (NCT04394975). Concurrently, it is important to assume that TKI monotherapy will lose significance for later lines of therapy, at least when considering patients being represented in clinical trials. Moreover, new therapeutic agents such as HIF-2α inhibitors (NCT03634540, NCT04586231, NCT04736706) or pan-HDAC inhibitors (NCT03592472) are also being tested in combination with TKI. For an overview of ongoing clinical trials containing TKI, see Table 2.

Given their current importance and potential future status in real-world mRCC therapy, knowing the characteristics and clinical application of various TKIs (see Table 3) to individually select THE best treatment will continue to play a major role for uro-oncologists.

## Figures and Tables

**Figure 1 cancers-14-03777-f001:**
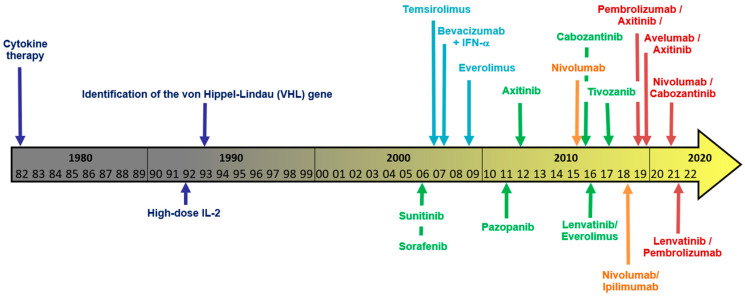
Evolution of treatment options for metastatic RCC—“from the dark to the golden age?”—timeline of mRCC treatments. Letter colours: Dark blue: Early milestones; green: TKI-containing regimens; orange: IO and IO–IO combination therapy; red: IO–TKI combination therapy; light blue: Other systemic therapies (non-TKI/non-IO).

**Table 1 cancers-14-03777-t001:** List of all relevant clinical studies on TKI monotherapy for mRCC with reported treatment efficacy and safety data.

	First Line				
Study name	**SUTENT**	**CABOSUN**	**AXIPAP**	**COMPARZ**	
Year	2009	2017	2020	2013	
N	750	157	44	1110	
N_groups	375/375	79/78	(13/30)	557/553	
Recruiting period	Aug 2004–Oct 2005	Jul 2013–Apr 2015	Oct 2015–Jan 2018	Aug 2008–Sep 2011	
Phase of study	3	2	2	3	
Intervention	sunitinib	cabozantinib	axitinib (type 1/type 2)	pazopanib	
Comparator	IFN-a	sunitinib	-	sunitinib	
Randomisation	1:1	1:1	-	1:1	
RCC subtype	m ccRCC	adv/m RCC, intermediate or poor risk	papillary RCC (type1/type 2)	m ccRCC	
Prior therapy	-	-	-	-	
Follow-up [years]		1.8	2.7	3.3	
mPFS_Interv. [months]	11	8.2	6.6 (6.7/6.2)	8.4	
mPFS_Comparator [months]	5	5.6		9.5	
mPFS_HR		0.66		1.05	
mPFS_p		0.012			
mOS_Interv. [months]	26.4	30.3	18.9 (not reached/17.4)	28.4	
mOS_Comparator [months]	21.8	21.8		29.3	
mOS_HR	0.821	0.80		0.91	
mOS_p	0.051	>0.05		0.28	
Objective response rate	47% vs. 12%	20% vs. 8%	28.6% (7.7%/35.7%)	33% vs. 29%	
Adverse Events		AE ≥ Gr. 3 66.7% vs. 68.1%	AE ≥ Gr. 3 54.5%	pazopanib < sunitinib (patient reported treatment side effects, *p* = 0.03)	
	**First/Second Line**				
Study name	**VEG105192**	**TIVO-1**	**SWOG 1500**		
Year	2010	2013	2021		
N	435	517	147		
N_groups	290/145	260/257	44 (/29 */28 *)/46		
Recruiting period	Apr 2006–Apr 2007	Feb 2010–Aug 2010	Apr 2016–Dec 2019		
Phase of study	3	3	2		
Intervention	pazopanib	tivozanib	cabozantinib (/savolitinib */crizotinib *)		
Comparator	placebo	sorafenib	sunitinib		
Randomisation	2:1	1:1	1(:1 *:1 *):1		
RCC subtype	adv/m ccRCC	m ccRCC with prior nephrectomy	papillary RCC (type1/type 2)		
Prior therapy	treatment-naive or cytokine therapy	-/one therapy line (excl. TKI or mTOR-I)	-/one therapy line (excl. VEGF-directed or MET-directed therapy))		
Follow-up [years]	3.8	1.6			
mPFS_Interv. [months]	9.2	11.9	9.0 (/3.0 */2.8 *)		
mPFS_Comparator [months]	4.2	9.1	5.6		
mPFS_HR	0.46	0.797	0.60		
mPFS_p	<0.001	0.042	0.02		
mOS_Interv. [months]	22.9	29.3	20.0 (/11.7 */19.9 *)		
mOS_Comparator [months]	20.5 (crossover allowed)	28.8	16.4		
mOS_HR	0.91	1.25	0.84		
mOS_p	0.224	0.105	>0.05		
Objective response rate	30% vs. 3%	33.1% vs. 23.1%	23% (/3%/0%) vs. 4%		
Adverse Events	AE ≥ Gr. 3 33% vs. 7%	AE ≥ Gr. 3 61% vs. 70%	AE ≥ Gr. 3 74% (39%/37%) vs. 69%		
	**Second Line**				**Third line**
Study name	**TARGET**	**METEOR**	**AXIS**	**(NCT01136733)**	**TIVO-3**
Year	2007	2016	2013	2015	2020
N	903	658	723	153	350
N_groups	452/451	330/328	361/362	51/52/50	175/175
Recruiting period	Nov 2003–Mar 2005	Aug 2013–Nov 2014	Sep 2008–Jul 2010	Mar 2012–Jun 2013	May 2016–Aug 2017
Phase of study	3	3	3	2	3
Intervention	sorafenib	cabozantinib	axitinib	Lenvatinib + everolimus	tivozanib
Comparator	placebo	everolimus	sorafenib	lenvatinib/everolimus	sorafenib
Randomisation	1:1	1:1	1:1	1:1:1	1:1
RCC subtype	m cc/ncc RCC post IFN-a	adv/m ccRCC post TKI	m ccRCC	adv/m ccRCC	m ncc/cc RCC
Prior therapy	IFN-a	TKI	not defined	VEGF-targeted therapy	≥two therapy lines, ≥one TKI
Follow-up [years]	2.0	1.5	3.0	2.0	1.6
mPFS_Interv. [months]	5.5	7.4	8.3	14.6	5.6
mPFS_Comparator [months]	2.8	3.9	5.7	7.4/5.5	3.9
mPFS_HR	0.51	0.51	0.66	0.40/0.61	0.73
mPFS_p	<0.001	<0.001	<0.001	<0.001/0.12	0.016
mOS_Interv. [months]	19.3	21.4	20.1	25.5	16.4
mOS_Comparator [months]	15.9	16.5	19.2	18.4/17.5	19.7
mOS_HR	0.77	0.66	0.969	0.55/0.74	0.99
mOS_p	0.02	<0.001	0.374	0.06/0.30	0.95
Objective response rate	10% vs. 2%	17% vs. 3%	23% vs. 12%	43% vs. 27%/6%	18% vs. 8%
Adverse Events	SAE 34% vs. 24%	AE ≥3 39% vs. 40%	treatment discontinuation due to toxic effects: 4% vs. 8%	AE ≥ Gr. 3 71% vs. 79%/50%	AE 84% vs. 94%, SAE 11% vs. 10%
	**Adjuvant**				
Study name	**S-TRAC**	**ASSURE**	**SORCE**	**PROTECT**	**ATLAS**
Year	2016	2016	2020	2017	2018
N	615	1943	1711	1538	724
N_groups	309/306	647/649/647	430/642/639	571 (p600)/564/198 (p800)/205	363/361
Recruiting period	Sep 2007–Apr 2011	Apr 2006–Sep 2010	Jul 2007–Apr 2013	Dec 2010–Sep 2013	May 2012–July 2016
Phase of study	3	3	3	3	3
Intervention	sunitinib (1 year)	sunitinib/sorafenib (1 year)	sorafenib (1year, +Placebo (2years))/sorafenib (3 years)	pazopanib (1year)	axitinib (1 year–3 years)
Comparator	placebo (1 year)	placebo (1 year)	placebo (3 years)	placebo (1 year)	placebo (1 year–3 years)
Randomisation	1:1	1:1:1	3:3:2	1:1	1:1
RCC subtype	nm ccRCC high risk (UISS criteria)	nm cc/ncc RCC high risk	nm cc/ncc RCC interm./high risk of recurrence	ccRCC, pT2 (high grade) or ≥ pT3 or pN+	nm ccRCC ≥ pT2 and/or N+
Prior therapy	-	-	-	-	-
Follow-up [years]	5.4	5.8	6.5	3.5 (p600)/4.0 (p800)	NA
mDFS_Interv. [years]	6.8	5.8/6.1	6.98/6.81	not reached	
mDFS_Comparator [years]	5.6	6.6	6.82	4.5	
mDFS_HR	0.76	1.02/0.97	0.94/1.01	0.8 (all)/0.69 (p800)/0.94 (p600)	0.87
mDFS_p	0.03	0.804/0.718	0.988	0.013/0.020 (p800)/0.51 (p600)	0.3211
mOS_Interv. [months]	not reached	not reached	not reached	not reached	not reached
mOS_Comparator [months]	not reached	not reached	not reached	not reached	not reached
mOS_HR	1.01	1.17/0.98	0.92/1.06	0.82 (all)/0.89 (p800)/0.79 (p600)	1.03
mOS_p	0.94	0.176/0.858		0.15/0.65/0.16	0.92
Adverse Events	AE ≥ 3 63.4% vs. 21.7%	AE ≥ 3 63%/72%/25%	AE ≥ 3 58.6%/63.9%/29.2%	AE ≥ 3 60% (P600)/66% (P800)/21%	AE ≥ 3 61% vs. 30%

* study groups were closed after interim analysis due to low efficacy; N: Number; RCC: Renal cell carcinoma; mPFS: Median progression-free survival; HR: Hazard ratio; mOS: Median overall survival; adv: Advanced; m: Metastatic; TKI: Tyrosine kinase inhibitor; ccRCC: Clear-cell renal cell carcinoma; mTOR-I: Mammalian target of rapamycin inhibitors; VEGF: Vascular endothelial growth factor; IFN-a: Interferone-alpha; AE: Adverse events; SAE: Severe adverse events; nm: Nonmetastatic; ncc: Non-clear-cell; p600: Pazopanib 600mg/d; p800: Pazopanib 800 mg/d.

**Table 2 cancers-14-03777-t002:** Ongoing trials including TKI for renal cell carcinoma. Trials with established therapies concerning setting and histopathological subtype were excluded, while phase I/II trials were excluded.

NCT Number	Name	Phase	Setting	Patient Group	Intervention	Comparator	Primary Endpoint	Estimated Enrollment	Estimated Study Completion	Status
NCT04995016		2	neoadjuvant	M0 ccRCC	pembrolizumab + axitinib	-	pathologic response	18	2023	Not yet recruiting
NCT04118855		2		M0 ccRCC	toripalimab + axitinib	-	ORR	30	2026	Not yet recruiting
NCT04022343		2		M0 ccRCC	cabozantinib	-	ORR	19	2023	Active, not recruiting
NCT03341845		2		ir/hr RCC	axitinib + avelumab	-	PRR	40	2025	Recruiting
NCT04393350		2		M0 RCC	lenvatinib + pembrolizumab	-	ORR	17	2024	Recruiting
NCT04370509		2		M0/M1 RCC	pembrolizumab/pembrolizumab + axitinib	-	TIIC	84	2025	Recruiting
NCT05172440		2		M0 ccRCC	axitinib + tislelizumab	-	ORR	20	2024	Active, not recruiting
NCT00715442		2		M1 RCC before CN	sunitinib	-	PFS	50	2022	Active, not recruiting
NCT05124431		2	inoperable/metastatic	nccRCC FL	anlotinib + everolimus	-	ORR	30	2024	Not yet recruiting
NCT04958473		2		recurrent/M1 RCC	sintilimab + axitinib	-	ORR	40	2025	Not yet recruiting
NCT05176288		2		M1 ccRCC	axitinib + avelumab + palbociclib	-	ORR	25	2023	Not yet recruiting
NCT04704219	KEYNOTE-B61	2		M1 nccRCC	pembrolizumab + lenvatinib	-	ORR	152	2025	Active, not recruiting
NCT04267120	LENKYN	2		LA/M1 nccRCC	pembrolizumab + lenvatinib	-	ORR	34	2027	Recruiting
NCT03967522	CABRAMET	2		M1 RCC with BN	cabozantinib	-	intracranial PFS	77	2024	Recruiting
NCT03562507		2		M1 RCC	ESK981 + nivolumab	-	ORR	28	2023	Active, not recruiting
NCT01217931	START	2		M1 RCC	sequential pazopanib/bevacizumab/everolimus (6 arms)	-	PFS	180	2023	Active, not recruiting
NCT05411081	PAPMET2	2		M1 PRCC	atezolizumab + cabozantinib	cabozantinib	PFS	180	2027	Not yet recruiting
NCT05048212		2		M1 RCC with BN FL	nivolumab + ipilimumab + cabozantinib	-	intracranial PFS	40	2024	Not yet recruiting
NCT05256472		2		M1 ccRCC FL	AK104 + axitinib	-	ORR	40	2024	Not yet recruiting
NCT02819596	CALYPSO	2		M1 RCC	savolitinib + durvalumab/savolitinib/durvalumab/durvalumab + tremelimumab	-	ORR	181	2022	Active, not recruiting
NCT05220267		2		LA/M1 nccRCC	anlotinib + sintilimab	-	PFS	43	2024	Not yet recruiting
NCT04904302		2		M1 ccRCC	sitravatinib + nivolumab	-	ORR, DCR	88	2023	Recruiting
NCT05012371		2		M1 RCC after IO	lenvatinib + everolimus	cabozantinib	PFS	90	2023	Recruiting
NCT01130519		2		M1 PRCC/HLRCC	bevacizumab + erlotinib	-	ORR	83	2023	Active, not recruiting
NCT05096390		2		LA/M1 PRCC FL	axitinib + pembrolizumab	axitinib	ORR	72	2025	Not yet recruiting
NCT03595124		2		M1 translocation RCC	axitinib + nivolumab	nivolumab	PFS	40	2031	Recruiting
NCT03092856		2		M1 RCC	PF-04518600 + axitinib	placebo + axitinib	PFS	104	2023	Recruiting
NCT03635892		2		M1 nccRCC	nivolumab + cabozantinib	-	ORR	97	5th July	Recruiting
NCT04071223	RadiCaL	2		RCC with bone metastasis	radium 223 + cabozantinib	cabozantinib	SSEFS	210	2024	Recruiting
NCT03634540		2		M1 ccRCC	belzutifan + cabozantinib	-	ORR	118	2025	Recruiting
NCT04413123		2		M1 nccRCC	nivolumab + ipilimumab, then nivolumab + cabozantinib	-	ORR	60	2024	Recruiting
NCT03685448	UNICAB	2		M1 nccRCC after IO	cabozantinib	-	ORR	48	2024	Recruiting
NCT04987203		3		M1 RCC after IO	tivozanib + nivolumab	tivozanib	PFS	326	2025	Recruiting
NCT04394975		3		M1 RCC	toripalimab + axitinib	sunitinib	PFS	380	2023	Recruiting
NCT03592472	RENAVIV	3		LA/M1 RCC	pazopanib + abexinostat	pazopanib + placebo	PFS	413	2022	Recruiting
NCT03937219	COSMIC-313	3		ir/hr RCC/M1 RCC	cabozantinib + nivolumab + ipilimumab	placebo + nivolumab + ipilimumab	PFS	840	2025	Active, not recruiting
NCT04523272		3		M1 RCC	TQB2450 + anlotinib	sunitinib	PFS	418	2023	Recruiting
NCT05043090	SAMETA	3		LA/M1 PRCC	savolitinib + durvalumab/durvalumab	sunitinib	PFS	220	2025	Recruiting
NCT04586231	MK-6482-011	3		M1 ccRCC after IO	belzutifan + lenvatinib	cabozantinib	PFS, OS	708	2024	Recruiting
NCT04338269	CONTACT-03	3		LA/M1 RCC after IO	atezolizumab + cabozantinib	cabozantinib	PFS, OS	523	2024	Active, not recruiting
NCT03793166	PDIGREE	3		M1 RCC FL	nivolumab + ipilimumab, then nivolumab + cabozantinib	nivolumab + ipilimumab, then nivolumab	OS	1046	2022	Recruiting
NCT04736706	MK-6482-012	3		M1 ccRCC FL	pembrolizumab + belzutifan + lenvatinib/pembrolizumab/quavonlimab + lenvatinib	pembrolizumab + lenvatinib	PFS, OS	1431	2026	Recruiting

M0: Non-metastatic; cc: clear-cell; ir/hr: Intermediate-/high-risk; ncc: Non-clear-cell; M1: Metastatic; CN: Cytoreductive nephrectomy; FL: First-line; LA: Locally advanced; IO: Immunotherapy; BN: Brain metastasis; PRCC: Papillary renal cell carcinoma; HLRCC: Hereditary leiomyomatosis and renal cell carcinoma; TKI: Tyrosine kinase inhibitor; PRR: Partial response rate; TIIC: Tumour-infiltrating immune cells; SSEFS: Symptomatic skeletal event (SSE)-free survival.

**Table 3 cancers-14-03777-t003:** List of all tyrosine kinase inhibitors applied to treat renal cell carcinoma, their dosage, and targets.

	Mono-Therapy	Combined-Therapy	Target	Further Indications
Sunitinib	60 mg once daily; dose reduction/increase by 12.5 mg possible (min. 12.5 mg; max 75 mg):	-	c-Kit, VEGFR1-3, PDGFR-α, PDGFR-β, FLT3, CSF-1R, RET	Gastrointestinal stromal tumours, Pancreatic neuroendocrine tumours,
Sorafenib	400mg twice daily; reduction to 200mg twice daily or 200/day	-	VEGFR2, FLT3, PDGFR, FGFR1	Hepatocellular carcinoma, differentiated thyroid carcinoma
Axitinib	5 mg twice daily; dose reduction: 2 × 3 mg and 2 × 2 mg; dose increase: 2 × 7mg and 2 × 10 mg	5 mg twice daily; dose reduction:2 × 3 mg and 2 × 2 mg; dose increase: 2 × 7 mg and 2 × 10 mg in combination with Pembrolizumab or Avelumab	VEGFR1-3	-
Tivozanib	1340 mg once daily; Dose reduction: 890 mg	-	VEGFR1-3, PDGFR-α/β, c-Kit, Tie2, ephb2	-
Cabozantinib	60 mg once daily; dose reduction 40 mg/20 mg	40 mg once daily in combination with Nivolumab	ET, MET, VEGFR-1-3, KIT, TRKB, FLT-3, AXL, TIE-2	Hepatocellular carcinoma, differentiated thyroid carcinoma
Pazopanib	800 mg once daily; reduced dosage by 200 mg until 200mg once daily possible	-	VEGFR1-3, P PDGFR-α/β, FGFR1/3, c-kit	soft-tissue sarcoma
Lenvatinib	-	20mg once daily in combination with Pembrolizumab; 18mg once daily in combination with Everolimus	VEGFR, PDGFRa, KIT; RET, FGFR	Hepatocellular carcinoma, differentiated thyroid carcinoma, Endometrial Cancer

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
