# Peer review of "Tyrosine Kinase Inhibitors in the Treatment of Metastasised Renal Cell Carcinoma—Future or the Past?"

_cancers, 2022, doi:10.3390/cancers14153777_

Round 1

Reviewer 1 Report

Dear Editor, thank you so much for inviting me to revise this manuscript about renal cell carcinoma.

This study addresses a current topic.

The manuscript is quite well written and organized. English could be improved.

Figures and tables are comprehensive and clear.

The introduction explains in a clear and coherent manner the background of this study.

We suggest the following modifications:

·      Introduction section: although the authors correctly included important papers in this setting, we believe the changing scenario of medical treatment for RCC should be better discussed and some recently published studies should be cited within the introduction ( PMID: 34894318; PMID: 33714725; PMID: 35193819.), only for a matter of consistency. We think it might be useful to introduce the topic of this interesting study.

·      Sunitinib section: Very interesting and timely discussion. Of note, the authors should expand this part. Several years of treatment with sunitinib worldwide, just some words. Please modify accordingly.

In addition, the authors should include a more personal perspective to reflect on. For example, they could answer the following questions – in order to facilitate the understanding of this complex topic to readers: what potential does this study hold? What are the knowledge gaps and how do researchers tackle them? How do you see this area unfolding in the next 5 years? We think it would be extremely interesting for the readers.

However, we think the authors should be acknowledged for their work. In fact, they correctly addressed an important topic in renal cell carcinoma, the methods sound good and their discussion is well balanced.

One additional little flaw: the authors could better explain the limitations of their work, in the last part of the Discussion.

We believe this article is suitable for publication in the journal although major revisions are needed. The main strengths of this paper are that it addresses an interesting and very timely question and provides a clear answer, with some limitations.

We suggest a linguistic revision and the addition of some references for a matter of consistency. Moreover, the authors should better clarify some points.

Author Response

This study addresses a current topic.

The manuscript is quite well written and organized. English could be improved.

Figures and tables are comprehensive and clear.

The introduction explains in a clear and coherent manner the background of this study.

We thank reviewer 1 for their very helpful comments and kind revision of our manuscript. The comments of reviewer 1 are addressed point by point in the following section. We revised the manuscript thoroughly.

  1. Introduction section: although the authors correctly included important papers in this setting, we believe the changing scenario of medical treatment for RCC should be better discussed and some recently published studies should be cited within the introduction ( PMID: 34894318; PMID: 33714725; PMID: 35193819.), only for a matter of consistency. We think it might be useful to introduce the topic of this interesting study.

We thank reviewer one for pointing us towards these studies, which give additional value to our introduction section. The section was revised and these studies included.

  1. Sunitinib section: Very interesting and timely discussion. Of note, the authors should expand this part. Several years of treatment with sunitinib worldwide, just some words. Please modify accordingly.

The Sunitinib section was expanded to include more information about the clinical experience with sunitinib treatment.

  1. In addition, the authors should include a more personal perspective to reflect on. For example, they could answer the following questions – in order to facilitate the understanding of this complex topic to readers: what potential does this study hold? What are the knowledge gaps and how do researchers tackle them? How do you see this area unfolding in the next 5 years? We think it would be extremely interesting for the readers.

We thank reviewer 1 for pointing out these additional topics which are certainly of high interest to the reader. We expanded our discussion section including these questions.

  1. However, we think the authors should be acknowledged for their work. In fact, they correctly addressed an important topic in renal cell carcinoma, the methods sound good and their discussion is well balanced.One additional little flaw: the authors could better explain the limitations of their work, in the last part of the Discussion.

Limitations of the study were included in the discussion section as well.

We believe this article is suitable for publication in the journal although major revisions are needed. The main strengths of this paper are that it addresses an interesting and very timely question and provides a clear answer, with some limitations. We suggest a linguistic revision and the addition of some references for a matter of consistency. Moreover, the authors should better clarify some points.

We thank reviewer 1 again for their interest in our manuscript and constructive comments. We are confident, that our revised manuscript addressed all issues brought up by reviewer 1. The manuscript was revised by a native speaking editor prior to submission.

Reviewer 2 Report

The authors did a great job putting together the details of TKIs for mRCC. However, the following points need to be addressed:  - Line 98- typo, 'mulitkinase'.
- Also, add an additional column indicating the other cancers where the TKI is approved or under trial. This will assist the readers in understanding the nature of the TKI, i.e., whether it is broad and has been used for other cancers or is limited to RCC/mRCC only.
- A paragram about resistance and relapse to TKIs in RCC/mRCC is highly recommended (including resistance and relapse mechanisms).
- (Important) A table about the ongoing trials of TKIs in RCC/mRCC (monotherapy or in combination) having the following heading is recommended (Trial number, Study design, number of patients enrolled, Disease description, Treatment Primary endpoint). For the combination therapy of TKIs, starting line 437, a separate table will be beneficial. Both tables can be combined. 
- A couple of lines about comparing the new generation of TKIs in RCC with the previous ones is desirable, especially when it comes to tolerance of TKIs in RCC/mRCC. 

Author Response

The authors did a great job putting together the details of TKIs for mRCC.

However, the following points need to be addressed: 

We thank reviewer 2 for their interest in our manuscript and their kind comments. We are confident that we are able to answer all raised questions sufficiently. Please see below.

1.- Line 98- typo, 'mulitkinase'.

The typo was corrected, thank you for pointing this out.

2.- Also, add an additional column indicating the other cancers where the TKI is approved or under trial. This will assist the readers in understanding the nature of the TKI, i.e., whether it is broad and has been used for other cancers or is limited to RCC/mRCC only.

The additional column was added to table 2.

3.- A paragram about resistance and relapse to TKIs in RCC/mRCC is highly recommended (including resistance and relapse mechanisms).

This is an interesting point, thank you for pointing out this missing information in our manuscript. An additional paragraph was added to .

- (Important) A table about the ongoing trials of TKIs in RCC/mRCC (monotherapy or in combination) having the following heading is recommended (Trial number, Study design, number of patients enrolled, Disease description, Treatment Primary endpoint). For the combination therapy of TKIs, starting line 437, a separate table will be beneficial. Both tables can be combined.

We consider comparison of the different combination therapies an important and challenging issue. Since many authors already addressed this topic extensively and we were at risk of insufficiently lighten its complexity as a mere subtopic, we deliberately decided against focusing on it. We added an additional table containing the information of ongoing trials containing TKIs in RCC/mRCC.

- A couple of lines about comparing the new generation of TKIs in RCC with the previous ones is desirable, especially when it comes to tolerance of TKIs in RCC/mRCC.

We added this information together with the information about treatment resistance in the new paragraph mentioned above.

Reviewer 3 Report

Michaelis et al enclosed a detailed review on tyrosine kinase inhibitors for metastasized renal cell carcinoma. The review should be of interest to the scientific community, however, the points below should be addressed:

1. The introduction is too short and does not give the sufficient background. Also, additional references need to be included in the introduction.

2. The text in Table 1 is too small and can not be easily read. This table needs to be reformatted and represented.

3. Minimal figures are included in the paper. Additional illustrative figures are needed

Author Response

Michaelis et al enclosed a detailed review on tyrosine kinase inhibitors for metastasized renal cell carcinoma. The review should be of interest to the scientific community, however, the points below should be addressed:

We thank reviewer 3 for their kind comments.

  1. The introduction is too short and does not give the sufficient background. Also, additional references need to be included in the introduction.

We revised the introduction thoroughly to give more background information. Additional references were added.

  1. The text in Table 1 is too small and cannot be easily read. This table needs to be reformatted and represented.

We reformatted table 1, separating the information. We assume that the final editing for publishing by the publisher can display the table on 2 pages making it easy for the reader.

  1. Minimal figures are included in the paper. Additional illustrative figures are needed.

We thank reviewer 3 for their valuable comments. After our revision, the manuscript includes now additional tables as well as added information. We believe in figures for illustrative purposes as well as to give additional information to the reader not conveyed within the text. After discussing this with all authors of this manuscript, we do not see any figure, which could deliver a certain purpose other than adding more “color” to the manuscript. We therefore did not follow reviewer 3 suggestion regarding this point and did not include additional figures. However, if reviewer 3 has a certain illustration in mind, that will be necessary to be included in the manuscript, we are happy to do another revision regarding this issue.

Round 2

Reviewer 1 Report

The authors should include the new references in the specific paragraph and section, and not in the text as PMID (as reported in the Introduction). 

Please revise accordingly.

Author Response

We changed the PMIDs to proper references, we ask reviewer 1 to forgive this mistake, it was simply forgotten to reformat the manuscript in the 1. revision.

Reviewer 2 Report

The authors addressed my points satisfactorily, and the manuscript can now be considered for publication!

Author Response

We thank reviewer 2

Round 3

Reviewer 1 Report

Acceptance.